# A compassion-based program to reduce psychological distress in medical students: A pilot randomized clinical trial

**Blanca Rojas[1], Elena Catalan[1,2], Gustavo Diez[3], Pablo Roca[4,5] \***

1 Medical School, Complutense University of Madrid, Madrid, Spain, 2 Virgen de la Victoria Hospital, Málaga, Spain, 3 Nirakara Lab, Complutense University of Madrid, Madrid, Spain, 4 Faculty of Health Sciences, Universidad Villanueva, Madrid, Spain, 5 Valencian International University, Valencia, Spain

\* pablo.roca@villanueva.edu

**Data Availability Statement:** The data are already available at https://github.com/nirakara-lab/CCT-medical-students.

## Abstract

### Objectives

Physicians and medical students are subject to higher levels of psychological distress than the general population. These challenges have a negative impact in medical practice, leading to uncompassionate care. This pilot study aims to examine the feasibility of Compassion Cultivation Training (CCT) to reduce psychological distress and improve the well-being of medical students. We hypothesize that the CCT program, as compared to a waitlist control group, will reduce psychological distress (i.e., stress, anxiety, and depression) and burnout symptoms, while improving compassion, empathy, mindfulness, resilience, psychological well-being, and emotion-regulation strategies after the intervention. Furthermore, we hypothesize that these improvements will be maintained at a two-month follow-up.

### Methods

Medical students were randomly assigned to an 8-week CCT or a Waitlist control group (WL). They completed self-report assessments at pre-intervention, post-intervention, and a 2-month follow-up. The outcomes measured were compassion, empathy, mindfulness, well-being, resilience, emotional regulation, psychological distress, burnout, and COVID-19 concern. Mixed-effects models and Reliable Change Index were computed.

### Results

Compared with WL, CCT showed significant improvements in self-compassion, mindfulness, and emotion regulation, as well as a significant decrease in stress, anxiety, and emotional exhaustion component of burnout. Furthermore, some of these effects persisted at follow-up. No adverse effects of meditation practices were found.

### Conclusions

CCT enhanced compassion skills while reducing psychological distress in medical students, this being critical to preserving the mental health of physicians while promoting

**Funding:** The author(s) received no specific funding for this work.

**Competing interests:** NO authors have competing interests.

compassionate care for patients. The need for institutions to include this type of training is also discussed.

## Introduction

The practice of medicine and the training to become a physician are associated with high levels of stress (e.g. patient death, treatment failures, difficult interactions with patients and families, and excessive cognitive demands) [1, 2]. Constant stress experienced by physicians can lead to psychological distress, attrition, and burnout syndrome [3, 4]. A striking concern is that the aforementioned problems may start in medical schools. Medical students are often overloaded with academic activities and burdened with constant stress and a sense of competition [5]. Furthermore, they are exposed, unlike other college students, to the particular stress associated with caring for others. As a result, medical students often experience high levels of psychological distress [6] and higher rates of depression, burnout, and even suicide, than in the general population [7].

Psychological distress harms not only individuals' well-being but also impedes their professional performance. Anxiety and burnout can increase medical errors [8], decrease empathy and thus foment dehumanization [9], diminish compassionate care [10], and lead to suboptimal patient outcomes [11]. Similarly, during their training, medical students become more frustrated and emotional distanced from patients, undergoing rising levels of cynicism, and becoming less empathic and compassionate [12], which in turn may degrade caretaking and optimal medicine practice [13, 14].

In view of the harmful effects of medical training and practice on both the practitioner's mental health as well as professional performance, the implementation of strategies and interventions to ameliorate the situation is vital. An alternative would be the practice of compassion. Several medical associations include compassion in their codes of ethics [15, 16], underscoring the importance of compassion [17] and compassionate patient care [18], and thus recognizing compassion as a pillar of medicine.

A growing body of evidence associates compassionate care with beneficial outcomes not only for patients [19] and doctor-patient relationships [20], but also for health-care providers and administrative institutions [21]. In fact, patients, families, clinicians, and policy makers consider compassion to have a greater healing effect than expertise alone [22], viewing compassion as the foundation of quality healthcare [22, 23]. However, compassion is a crosscutting competency in the medical curriculum, and it is usually not specifically addressed in any subjects or training.

Fortunately, evidence from the so-called Compassion-Based Interventions (CBIs) suggests that CBIs improve mindfulness, self-compassion and well-being, while reducing anxiety and depression symptoms [24, 25]. Also, in contrast with Mindfulness-Based Interventions (MBIs), CBIs have shown larger socio-emotional changes (i.e. common humanity and empathic concern) [26, 27], core skills in compassionate clinical practice. Furthermore, a recent literature review has shown that CBIs can increase physician empathy and compassion, medical students constituting the population with highest success rates [19].

Among the CBIs, the Compassion Cultivation Training (CCT) [28] is one of the programs that has shown promising results in recent years [26, 27, 29–31]: increases in compassion (both for self and others), empathy and mindfulness skills, reductions of psychological distress (stress, anxiety and depression) and burnout, improvements in well-being and emotion regulation, among others. Furthermore, the CCT program has also shown promising results in

medical-school students, helping students to address major stress associated with academic and clinical responsibilities [32].

Most CCT studies have focused on the general population, with few examples in healthcare workers [31, 33]. Only one study available has examined the application of CCT on medical students [32], conducting a qualitative analysis of self-reported experiences after the program without a comparison group or longitudinal measures. The present study aims to examine the effects of CCT on psychological outcomes in a sample of medical students. We hypothesize that the CCT program, as compared to a waitlist control group, will reduce psychological distress (i.e., stress, anxiety, and depression) and burnout symptoms, while improving compassion, empathy, mindfulness, resilience, psychological well-being, and emotion-regulation strategies after the intervention. Furthermore, we hypothesize that these improvements will be maintained at a two-month follow-up.

## Materials and methods

### Study design

The present study is a pilot two-arm randomized controlled trial, in compliance with CONSORT statements for pilot trials [34] (see S1 Table in S1 File for more details). Participants were randomly assigned in a 1:1 ratio to the Compassion Cultivation Training or the Waitlist control group. Randomization was performed using a Random Number Generation Function and was blind to the data scientist. Furthermore, randomization was performed after the baseline assessment to preserve adequate allocation concealment. Participants assigned to CCT were assessed before the intervention, at the end of the intervention, and in a 2-month follow-up. Participants assigned to WL were assessed at the same time points: pre-waitlist assessment, post-waitlist assessment, and follow-up-waitlist assessment. Participants in the WL received the CCT program following the follow-up-waitlist assessment.

Participation in the study was voluntary and participants gave their written informed consent prior to their inclusion in the study (all were of legal age). Recruitment was carried out between December 2020 and September 2021. The study protocol was approved by the hospital ethics committee prior to participant recruitment (Ref. 20/742-EC_X) and was conducted in compliance with the Declaration of Helsinki, and good clinical practice. Furthermore, the trial was pre-registered at ClinicalTrial.org (ID: NCT04690452).

### Participants

A total of 44 medical students (first to sixth year university students) were randomly assigned to an 8-week CCT program or WL. We used the G *Power (v. 3.1) to estimate sample size *a priori* to test mixed models (i.e., Group as within-subjects factor and Time as between-subjects factor). With a medium effect size of 0.40 [26, 30], and an alpha of .05, we would need at least 44 participants to detect significant effects at 95% power. We estimated a sample loss of less than 5%, and the final attrition rate was 5.5% in CCT and none for the WL.

Fig 1 illustrates the participation flow diagram. Four participants assigned to CCT withdrew before starting the intervention due to incompatibilities in their academic calendar. Finally, we analyzed the data of 40 participants: 18 participants in CCT group, and 22 in WL group. The participants, of which 60% were single and 92.5% were women, had a mean age of 23.4 ($SD$ = 5.59). The CCT and WL groups did not differ at baseline in age ($t_{(38)}$ = -.27, $p$ = .79), gender ($\chi^2_{(1)}$ = .18, $p$ = .67), nationality ($\chi^2_{(1)}$ = .18, $p$ = .67), marital status ($\chi^2_{(1)}$ = 1.36, $p$ = .24), prior meditation experience ($\chi^2_{(1)}$ = .04, $p$ = .84), or weekly formal meditation ($t_{(12)}$ = 1.17, $p$ = .27).

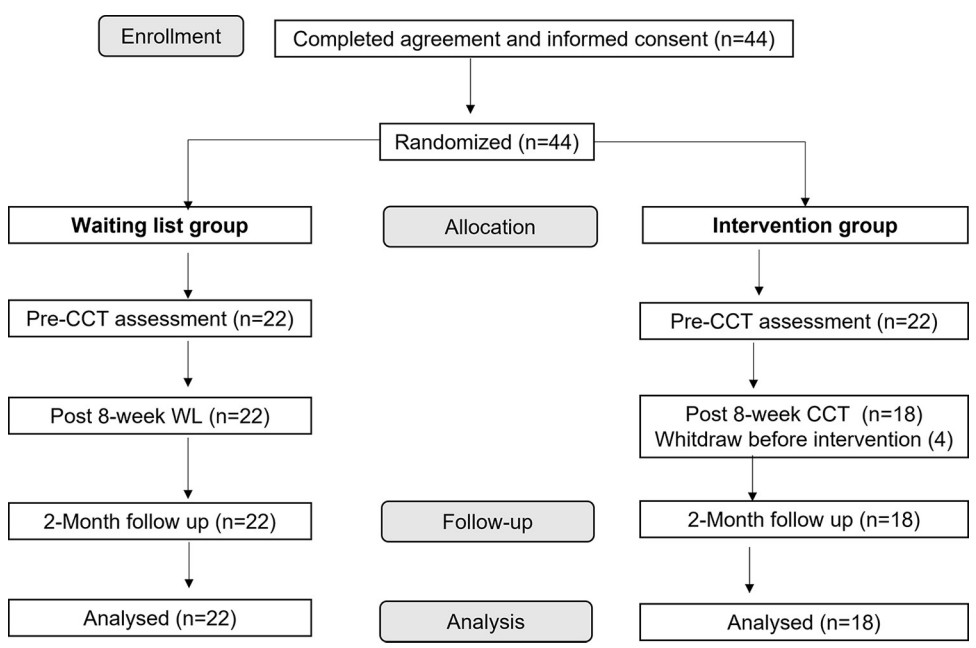

**Fig 1. Study CONSORT diagram.** *Note.* CCT: Compassion Cultivation Training; WL: Waitlist.

Eligibility criteria were: 1) being ≥ 18 years of age; 2) being enrolled as a medical student during the academic year 2020–2021; 3) having language fluency to understand the program and the evaluation; 4) providing written informed consent; 5) having internet access to attend the CCT sessions and online assessments; and 6) making a commitment to program attendance. Participants were excluded if they: 1) were diagnosed with a severe mental disorder in the active phase (e.g., schizophrenia, bipolar disorder); 2) were under the influence of alcohol and other substances during the weekly sessions (determined by the CCT instructor); 3) withdrew before starting the intervention; or 4) participated in any other meditation program during the study period.

## Procedure

Participants were invited to join the study at the beginning of the academic year (September 2020) on the university's official website. Eligible individuals received information about the study and gave their written informed consent prior to their inclusion. Then, all participants completed online assessments during the week before starting the program (i.e., pre-intervention assessment), during the week after the completion of the program (i.e., post-intervention assessment), and in the two-month follow-up (i.e., follow-up assessment). Afterwards, participants were randomized to either CCT or WL. The CCT was administered online (in a group format) due to the social-distancing measures during the COVID-19 pandemic. Participants received course credits and a book as a reward for their participation in the study.

## Intervention description

CCT is an 8-week standardized, secular, meditation program aimed at cultivating compassion and empathy toward oneself and others, which in turn reduces psychological distress and promotes well-being. CCT was developed by Geshe Thupten Jimpa in collaboration with a group of clinical psychologists and scientists supported by the Centre for Compassion and Altruism

Research and Education at Stanford University [28]. The CCT aimed at cultivating compassion and empathy toward oneself and others, including insights and techniques from psychology, neuroscience, and contemplative practice. The program was conducted in a group format and consists of weekly 2-hour online sessions. The sessions combined pedagogical instruction with active group discussions, together with in-class practical exercises, and guided group meditations. In addition, participants engaged in 30 minutes of daily home practice (i.e., formal practice), as well as real-world assignments for practicing compassionate thoughts and actions (i.e., informal practice). At the end of each session, participants received a set of pre-recorded audio files and a workbook to help with their daily practices. One of the authors (BR), a medical doctor certified as instructor by the Compassion Institute, taught the program. The CCT program comprises six sequential steps (see Table 1): 1) learning to focus and settle the mind; 2) experiencing compassion and loving-kindness for a loved one; 3) experiencing compassion and loving-kindness for oneself; 4) experiencing compassion towards others, premised in common humanity and interconnectedness; 5) experiencing compassion towards all beings; and 6) "active compassion" practice (*Tonglen*).

## Measures

The online assessment included a set of scales evaluating different domains related to compassion practice: [26, 30, 31, 35–37] compassion (both for others and oneself), empathy,

**Table 1. Compassion Cultivation Training protocol.**

| Week | Contents |
|---|---|
| 1 | **Step 1.** Settling and focusing the mind |
| | Introduction of basic skills to still and focus the mind through breath focused meditation. This step is considered foundational for all subsequent practices in the program. |
| 2 | **Step 2**. Loving-kindness and compassion for a loved one |
| | Learning to recognize how the experiences of love and compassion feel when they occur naturally. Meditation and practical exercises aim to help practitioners recognize the physical and physiological feelings of warmth, tenderness, concern, and compassion. |
| 3 | **Step 3a.** Compassion for oneself. |
| | Learning to develop self-acceptance, tenderness, non-judgment, and caring in self-to-self relations. Connecting with one's own feelings and needs and relating them with compassion is the basis for developing a compassionate stance toward others. |
| 4 | **Step 3b.** Loving-kindness for oneself. |
| | Learning to develop qualities of warmth, appreciation, joy, and gratitude in self-to-self relations. While the previous step focused on self-acceptance, this step focuses on developing appreciation for oneself. |
| 5 | **Step 4.** Embracing shared common humanity and developing appreciation of others |
| | Establishing the basis for compassion toward others by recognizing our shared common humanity. Appreciating the kindness of others and how human beings are deeply interconnected. |
| 6 | **Step 5**. Cultivating compassion for others |
| | Based on the previous step, participants begin to cultivate compassion for all beings by moving progressively from focusing on a loved one, to a neutral person, difficult person, and finally, all beings. |
| 7 | **Step 6.** Active compassion practice |
| | This step explicitly evokes the altruistic wish to alleviate others' suffering. This involves a visualization practice where the practitioner imagines taking away the suffering of others and giving them what is beneficial in oneself. This practice is known as tonglen or "giving and taking" in Tibetan Buddhism. |
| 8 | **Integrated practice** |
| | The core elements of all six steps are combined into an "integrated compassion meditation practice" that can be practiced daily by participants who choose to adopt it. |

Note. Adapted from Brito-Pons, Campos & Cebolla, 2018.

**Table 2. Summary of measures.**

| Changes measured | Scale used for assesement | α |
|---|---|---|
| **Primary outcomes** | | |
| Compassion to others | Compassion Scale Pommier (CSP) [38], 24-item [39]. | .83 |
| Self-compassion | Self-Compassion Scale, Short Form (SCS-SF) [39], 12-item | .90 |
| Empathy | Interpersonal Reactivity Index (IRI) [40], 28-item. | .76 |
| Psychological distress | Depression Anxiety Stress Scales (DASS-21) [41], 21-item. | .85 |
| General well-being | Pemberton Happiness Index (PHI) [42], 11-item. | .95 |
| **Secondary outcomes** | | |
| Mindfulness | Five-Facet Mindfulness Quest Short (FFMQ) [43], 20-item. | .83 |
| Burnout | Maslach Burnout Inventory-Student (MBI-SS) [44], 15-item. | .75 |
| Emotion Regulation | Difficulties in Emotion Regulation Scale (DERS) [45], 28 item. | .95 |
| Resilience | Brief Resilience Scale (BRS) [46], 5-item. | .86 |
| Concerns COVID-19 | A single item. | - |
| Acceptability/ satisfaction | Adaptation of Based on Mindfulness-based Teaching Assessment Criteria (MBI-TAC) [47] and home practice. | - |

α: Cronbach's α (reliability) found in the current study

Mindfulness, psychological well-being, resilience, psychological distress, burnout, and concerns about COVID-19. Furthermore, we also assessed program acceptability and satisfaction. Table 2 offers a brief description of the scales, as well as the reliability found in this study (see S1 File).

## Data analysis

Chi-square test for categorical data and independent Student's *t* test for continuous data were performed to confirm that there were no baseline differences between CCT and WL groups. Missing data were also explored, revealing that there were only 6.1% of overall missing values completely at random (Little MCAR test: $\chi^2_{(632)}$ = 225,265, *p* = .99). Only one drop-out case was found in the CCT group. Given that imputation of missing values is not necessary before performing longitudinal mixed-model analysis [48], we did not impute the missing data.

Mixed-effects models were conducted to analyze the effects of the CCT program, using the *lmer* function from the *lme4* R-package [49]. R version 4.0.2 was used for the analyses [50]. Analyses were conducted via Restricted Maximum Likelihood estimation (REML) [51, 52], which provides a less-biased estimate of variance components with smaller sample sizes and missing data [51, 53, 54]. Variance across participants was modelled as a random effect in the model to account for individual differences in the dependent variable. Group (i.e., CCT vs WL) and Time (i.e., pre, post, and 2-month follow-up) were modelled as fixed effects. We used the WL and pre-intervention as reference categories in the analysis. Fixed-effect parameters were interpreted as the regression weights in the linear regression models [55], in which parameter estimates reflect changes in the mean of the dependent variable between the contrast and reference groups. Furthermore, the effect sizes of each model were presented as the model-derived fixed-effect parameter regression weights [56]. Tukey-corrected post hoc comparisons were computed to determine which interactions are responsible for the significant differences.

To improve individual-level analysis and the detection of potential adverse effects of the intervention, we also computed the Reliable Change Index (RCI) [57, 58] on pre-post scores

for the main clinical outcomes (i.e., stress, anxiety, depression, emotional exhaustion, and cynicism). Participants were classified into four different categories based on their cut-off and RCI scores: (1) No change: when post-intervention scores did not reach the functional cut-off and the change was not reliable; (2) Improved: when the change was reliable but post-intervention scores did not reach the functional cut-off level; (3) Recovered: when post-intervention scores were located within the range of the functional cut-off distribution and the change was reliable; and 4) Deteriorated: when post-intervention scores were worse than the pre-intervention.

## Results

Fig 2 shows fixed-effect parameter estimates, and their corresponding 95% confidence interdentals for each dependent variable (see also S1 Fig in S1 File).

### Compassion measures

Significant differences were found between CCT and WL in self-kindness, common humanity, and mindful self-compassion after the program was completed (i.e., a significant Group x Time2 interaction). Tukey-corrected *post hoc* comparisons showed a significant increase of these measures after the CCT (i.e., self-kindness $p = .003$; common humanity $p < .001$; mindful self-compassion $p = .005$), whereas no changes were found in WL (i.e., self-kindness $p = .884$; common humanity $p = .840$; mindful self-compassion $p = .962$). Furthermore, significant differences also appeared between CCT and WL in self-kindness at follow-up (i.e., a significant Group x Time3 interaction). *Post hoc* comparisons showed that improvements in self-kindness in CCT persisted at follow-up ($p < .001$), whereas no changes were detected in WL ($p = .847$). However, no significant differences between groups were found in compassion to others and empathic concern.

### Psychological distress and burnout measures

Significant differences were found between CCT and WL after the program in all the measures evaluated (i.e., a significant Group x Time2 interaction for stress, depression, anxiety, and emotional exhaustion, and cynicism). *Post hoc* comparisons indicated significantly lower

| | Group 1 | Time 2 | Time 3 | Group1 x Time 2 | Group 1 x Time 3 |
|---|---|---|---|---|---|
| **Mindfulness** | | | | | |
| Observing | b= .58 (.24), CI [.12, 1.05] | b= -.05 (.12), CI [-.28, .17] | b= .05 (.12), CI [-.18, .27] | b= .48 (.17), CI [.14, .81] | b= .38 (.17), CI [.04, .71] |
| Describing | b= .10 (.29), CI [-.46, .66] | b= -.07 (.12), CI [-.31, .17] | b= .02 (.12), CI [-.22, .26] | b= .31 (.18), CI [-.04, .66] | b= .29 (.18), CI [-.06, .65] |
| Acting with awareness | b= -.08 (.29), CI [-.64, .48] | b= .01 (.15), CI [-.28, .31] | b= .02 (.15), CI [-.27, .32] | b= -.37 (.23), CI [-.07, .8] | b= -.37 (.23), CI [-.06, .81] |
| Non-judging | b= -.10 (.3), CI [-.67, .47] | b= .05 (.15), CI [-.25, .34] | b= -.17 (.15), CI [-.13, .47] | b= .65 (.23), CI [.20, 1.09] | b= .37 (.23), CI [-.07, .81] |
| Non-reactivity | b= -.18 (.23), CI [-.62, .26] | b= -.37 (.13), CI [-.29, .21] | b= .06 (.23), CI [-.19, .31] | b= .62 (.19), CI [.25, .99] | b= .48 (.19), CI [.11, .85] |
| **Compassion** | | | | | |
| Self-Kindness | b= .01 (.31), CI [-.58, .61] | b= -.09 (.18), CI [-.44, .27] | b= -.10 (.23), CI [-.62, .26] | b= .76 (.27), CI [.24, 1.29] | b= .87 (.27), CI [.33, 1.40] |
| Common Humanity | b= -.01 (.28), CI [-.55, .52] | b= .09 (.17), CI [-.23, .41] | b= .15 (.17), CI [-.18, .47] | b= .61 (.25), CI [.14, 1.09] | b= .44 (.25), CI [-.04, .92] |
| Mindful self-compassion | b= -.22 (.31), CI [-.83, .39] | b= -.04 (.15), CI [-.33, .25] | b= -.04 (.15), CI [-.34, .25] | b= .57 (.22), CI [.14, 1.00] | b= .43 (.22), CI [-.00, .86] |
| Compassion to others | b= .02 (.13), CI [-.23, .27] | b= -.03 (.05), CI [-.13, .08] | b= -.09 (.06), CI [-.20, .01] | b= .04 (.08), CI [-.12, .19] | b= .09 (.08), CI [-.07, .25] |
| Empathic Concern | b= -.04 (.16), CI [-.35, .27] | b= -.03 (.09), CI [-.21, .15] | b= -.05 (.09), CI [-.23, .13] | b= .25 (.14), CI [-.02, .08] | b= -.18 (.14), CI [-.08, .45] |
| **Psychological distress** | | | | | |
| Stress | b= 1.38 (1.03), CI [-.61, 3.37] | b= -.28 (.81), CI [-1.85, 1.29] | b= .12 (.83), CI [-1.48, 1.71] | b= -2.83 (1.20), CI [-5.15, -.50] | b= -1.34 (1.21), CI [-3.69, 1.00] |
| Depression | b= .13 (1.34), CI [-2.46, 2.72] | b= 1.07 (.88), CI [-.63, 2.77] | b= -.06 (.90), CI [-1.79, 1.67] | b= -3.43 (1.30), CI [-5.96, -3.91] | b= -1.37 (1.31), CI [-3.91, 1.17] |
| Anxiety | b= 1.38 (1.15), CI [-.84, 3.59] | b= 1.03 (.79), CI [-.49, 2.56] | b= .82 (.80), CI [-.73, 2.37] | b= -3.94 (1.17), CI [-6.19, -1.68] | b= -3.31 (1.18), CI [-5.58, -1.04] |
| Emotional exhaustion | b= 1.70 (1.91), CI [-2.01, 5.40] | b= 1.08 (1.01), CI [-.85, 3.04] | b= -.67 (1.02), CI [-2.64, 1.32] | b= -6.07 (1.49), CI [-8.97, -3.20] | b= -4.08 (1.50), CI [-7.00, -1.19] |
| Cynicism | b= 3.83 (2.00), CI [-.05, 7.72] | b= 1.33 (.90), CI [-.41, 3.07] | b= 1.01 (.92), CI [-.76, 2.78] | b= -2.92 (1.33), CI [-5.51, -.35] | b= -2.72 (1.35), CI [-5.32, -0.12] |
| **Emotion regulation** | | | | | |
| Emotional inattention | b= .04 (1.10), CI [-2.09, 2.17] | b= 1.15 (.64), CI [-.08, 2.38] | b= 1.40 (.65), CI [.16, 2.66] | b= -2.64 (.94), CI [-4.46, -.82] | b= -2.49 (.95), CI [-4.32, -.65] |
| Emotional confusion | b= 1.06 (.89), CI [-.66, 2.78] | b= .40 (.55), CI [-.65, 1.47] | b= .57 (.56), CI [-.50, 1.65] | b= -2.98 (.81), CI [-4.55, -1.42] | b= -2.45 (.82), CI [-4.03, -.87] |
| Emotional rejection | b= .74 (2.62), CI [-4.35, 5.83] | b= -1.22 (1.47), CI [-4.05, 1.64] | b= -0.88 (1.50), CI [-3.78, 2.02] | b= -5.39 (2.18), CI [-9.60, -1.18] | b= -4.20 (2.20), CI [-8.44, .06] |
| Life interference | b= .15 (1.47), CI [-2.70, 3.00] | b= -.94 (.79), CI [-2.47, .60] | b= -.46 (.81), CI [-2.47, 1.10] | b= -2.08 (1.18), CI [-4.35, .19] | b= -.38 (1.19), CI [-2.67, 1.92] |
| Emotional lack of control | b= 2.59 (2.75), CI [-2.76, 7.93] | b= -.31 (1.51), CI [-3.22, 2.6] | b= .06 (1.54), CI [-2.91, 3.04] | b= -6.62 (2.24), CI [-10.94, -2.3] | b= -5.34 (2.26), CI [-9.69, -.98] |
| **Psychological well-being** | | | | | |
| Well-being | b= 4.25 (5.92), CI [-7.26, 15.7] | b= 3.02 (2.58), CI [-1.80, 8.18] | b= 5.26 (2.63), CI [.17, 10.34] | b= 6.77 (3.82), CI [-.61, 14.15] | b= 2.48 (3.86), CI [-4.97, 9.93] |
| Resilience | b= .04 (.26), CI [-.47, .55] | b= .11 (.14), CI [-.16, .37] | b= .13 (.14), CI [-.14, .39] | b= -.00 (.20), CI [-.39, .39] | b= .19 (.20), CI [-.21, .58] |
| Academic effectiveness | b= .16 (2.11), CI [-3.95, 4.26] | b= .89 (1.12), CI [-1.29, 3.06] | b= .31 (1.15), CI [-1.89, 2.53] | b= -.39 (1.66), CI [-3.60, 2.82] | b= -.11 (1.68), CI [-3.36, 3.12] |

**Fig 2. Fixed-effect parameter estimates (standard error), and their corresponding 95% confidence interdentals for each dependent variable.**

scores in stress ($p < .002$), depression ($p = .041$), anxiety ($p = .003$), and emotional exhaustion ($p < .001$) after the CCT, whereas no changes were found in WL (i.e., stress $p = .935$; depression $p = .451$; anxiety $p = .395$; emotional exhaustion $p = .533$). Furthermore, significant differences also appeared between CCT and WL in anxiety, emotional exhaustion, and cynicism at follow-up (i.e., a significant Group x Time3 interaction). *Post-hoc* comparisons showed that the decreases in anxiety ($p = .013$) and emotional exhaustion ($p < .001$) in CCT group remained at follow up, whereas no changes were found in WL (i.e., anxiety $p = .567$; emotional exhaustion $p = .792$). The case of cynicism was atypical, because significant differences between groups appeared at baseline (where CCT participants showed significantly higher cynicism levels), but these differences disappeared after the intervention.

## Psychological well-being measures

No significant differences between groups were found in psychological well-being, resilience, or academic effectiveness (i.e., the Group x Time interactions were not significant).

## Mindfulness measures

Significant differences were found between CCT and WL in observing, non-reactivity to inner experience, and non-judging after the program (i.e., a significant Group x Time2 interaction). Tukey-corrected *post hoc* comparisons showed a significant increase of these facets after the CCT (i.e., observing $p = .004$; non-reactivity $p < .001$; non-judging $p < .001$), whereas no changes were found in WL (i.e., observing $p = .885$; non-reactivity $p = .954$; non-judging $p = .951$). In addition, CCT and WL significantly differed in observing and non-reactivity at follow-up (i.e., a significant Group x Time3 interaction). *Post hoc* comparisons showed that improvements in observing ($p = .004$) and non-reactivity to inner experience ($p < .001$) were maintained in the CCT group at follow-up, whereas no changes were found in WL (i.e., observing $p = .917$; non-reactivity $p = .873$). No significant differences between groups were found in describing and acting with awareness factors.

## Emotional-regulation measures

Significant differences appeared between CCT and WL after the program in emotional inattention, emotional confusion, emotional lack of control, and emotional rejection (i.e., a significant Group x Time2 interaction). Tukey-corrected *post hoc* comparisons indicated a significant decrease in all these non-adaptive emotional-regulation strategies after the CCT (i.e., inattention $p = .008$; confusion $p < .001$; lack of control $p < .001$; and rejection $p < .001$), whereas no changes were found in WL (i.e., inattention $p = .176$; confusion $p = .745$; lack of control $p = .077$; and rejection $p = .688$). Furthermore, significant differences were also detected between CCT and WL in emotional inattention, confusion, and lack of control at follow-up (i.e., a significant Group x Time3 interaction). *Post hoc* comparisons showed that decreases in emotional confusion ($p = .007$) and emotional lack of control ($p = .005$) in CCT persisted at follow-up, whereas no changes were found in WL (i.e., confusion $p = .561$; lack of control $p = .999$). However, no significant differences appeared between groups in emotional life interference.

Regarding concern about the impact of COVID-19, a significant Group x Time interaction was found at follow-up, but not after the CCT. No significant differences between groups were found after the intervention, but these differences emerged at follow-up, where the CCT significantly reduced concern about COVID-19 impact ($p < .001$), but no changes were found in WL ($p = .899$).

## Significance of clinical improvements

Fig 3 shows the RCI for the main clinical outcomes. The RCI analyses indicated significant differences between CCT and WL in clinical change regarding stress ($\chi^2_{(3)}$ = 7.98, $p$ = .04), where CCT showed higher recovery rates whereas WL showed a higher number of deteriorated participants. Similarly, there were significant differences between CCT and WL in clinical change in anxiety ($\chi^2_{(3)}$ = 11.03, $p$ = .01), where CCT showed higher recovery and improvement rates whereas WL showed a higher number of participants registering no change and deteriorated. However, no significant differences were found in depression symptoms ($\chi^2_{(3)}$ = 5.50, $p$ = .14). Regarding burnout measures, significant differences between CCT and WL in clinical change were found in emotional exhaustion ($\chi^2_{(3)}$ = 11.73, $p$ < .01) and cynicism ($\chi^2_{(3)}$ = 8.56, $p$ = .04),

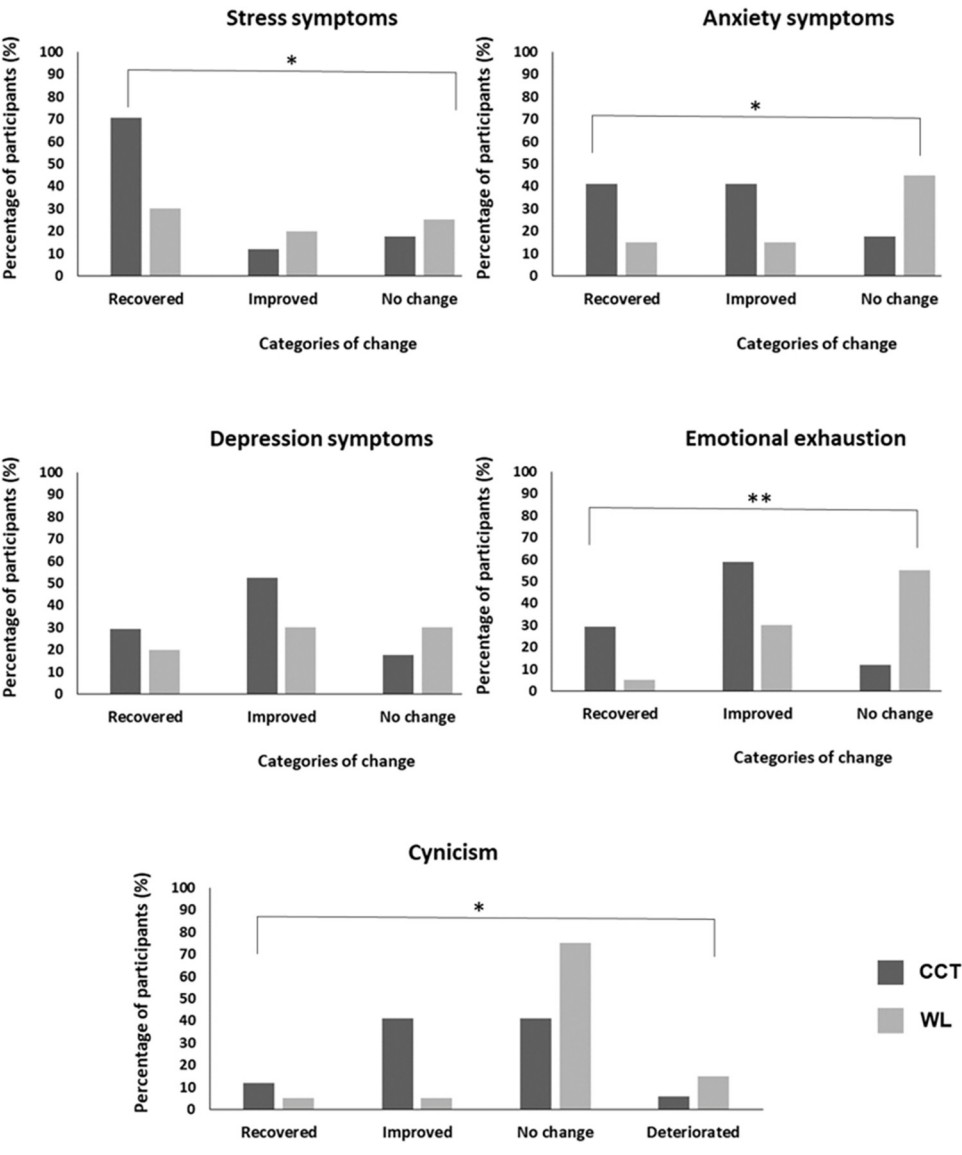

**Fig 3. Reliable Change Index for the main clinical outcomes.** *Note.* CCT: Compassion Cultivation Training; WL: Waitlist. *$p$ < 0.05; **$p$ < 0.01.

where CCT displayed a higher number of improved participants whereas WL showed a higher number of participants with no changes.

## Program acceptability and satisfaction

Regarding program satisfaction, all the participants stated that they would recommend the CCT to a friend. The average score of the participant's overall satisfaction was very high (rated 8.76 out of 10). A large proportion of the participants considered that the goals of the program had been reached (rated 8.11 out of 10), and they perceived strong relevance of the contents of each session to achieve the program goals (rated 9.36 out of 10). Participants were also very satisfied with the instructor's work and the materials provided, with average scores above 9. Furthermore, 94% of participants claimed they will continue practicing meditation after CCT had finished. In terms of perceived usefulness, participants found the meditations to be very helpful (8.1 out of 10). The most valued formal practices were attention to breathing ($M = 9$: $SD = 1.06$) and loving-kindness meditation ($M = 8.88$: $SD = 1.22$). Furthermore, participants perceived a high degree of applicability of the program's contents in their daily lives (rated 8.22 out of 10).

## Discussion

The mental health and well-being of physicians and medical students is a current issue of concern [5, 59]. Therefore, the aim of this study was to examine the effects of a Compassion-Based Intervention on psychological outcomes in medical students.

Based on previous studies [26, 27, 31, 36], we hypothesized that CCT would improve compassion levels, both for oneself and for others. Our results showed that CCT significantly improved self-compassion after the program (i.e., self-kindness, common humanity, and mindful self-compassion). The changes in self-compassion are broadly consistent with previous work in the field, both with CCT in general population [31, 60] and other CBIs [61], extending previous findings to a sample of medical students. The improvements in self-compassion in this population are crucial for several reasons. Firstly, improvements in awareness of their own suffering in the present moment could help them to regulate their negative thoughts and emotions in a preventive way [37, 62], helping them to reduce over-identification or to recognize their own psychological distress. Secondly, self-kindness would act as a protective factor against excessive self-criticism and self-judgment [61], helping them to accept that they are not perfect and will sometimes make mistakes in their medical work. Thirdly, common humanity improvements after the program would reduce the sense of isolation [26, 63], helping them to realize that suffering is part of the shared human experience. Furthermore, training self-compassion not only leads to improvements at a personal level, but could also enhance professional skills necessary for optimal medical practice [64].

In contrast to previous studies [26, 27, 36], we found no significant changes in compassion to others and empathic concern. The lack of changes in these variables might be explained by sample characteristics. For instance, medical students might be exerting a ceiling effect, having higher levels of compassion and empathy at baseline than in the general population. Furthermore, unlike physicians who have been practicing for years, medical students have not yet been exposed to large doses of daily suffering, so that their compassion levels to others are still preserved [65]. The fact that the intervention was delivered online during the COVID-19 pandemic (drastically reducing interpersonal interactions) could explain the lack of improvements in empathy and compassion to others.

The results of the present study also showed a significant decrease in psychological distress (i.e., stress, anxiety, and depression) after the CCT, whereas no changes were found in WL.

These results are in line with those found in previous studies [26, 27, 30, 66], showing that CCT is an effective intervention to improve mental health in the general population. Given the high level of psychological distress in physicians and medical students [6, 67], the positive effects of CCT on mental-health outcomes make it a promising program to be included in hospitals and medical school curriculum. The lack of maintenance of significant changes in stress and depression at follow-up may suggest the importance of introducing maintenance sessions throughout the academic year to sustain the effects of intervention and to support continued meditation practice.

Furthermore, we also found a significant decrease of burnout symptoms (i.e., emotional exhaustion) after the CCT program. This reduction is consistent with earlier studies showing that MBIs reduce burnout symptoms in healthcare providers [68, 69], extending previous findings to CBIs. In fact, self-compassion and mindfulness changes might be important mechanisms of change that explain the effects of CCT on psychological distress [27] and burnout symptoms [70–72].

Contrary to the results of previous studies [26, 31, 66], we found no significant changes in well-being variables after the CCT (i.e., psychological well-being, resilience, and academic effectiveness). A plausible explanation could be that CCT is focused mainly on "suffering". In the CCT, compassion is defined as the feeling that arises in witnessing the suffering of others and oneself, prompting a desire to reduce this suffering. Therefore, the program teaches skills to approach this suffering in a healthier and less avoidant way, but it does not have modules specifically aimed at enhancing positive affect. Importantly, participants showed high levels of satisfaction with the program, all of them would recommend the program to others, they perceived a high applicability of the program's contents in their daily lives, and most of them expressed intentions to continue practicing compassion meditation after the intervention.

Consistent with our hypothesis, mindfulness skills significantly improved after CCT (specifically the facets of observing, non-reactivity to inner experience, and non-judging of inner experience). Similar results have been found in previous studies in the field [30, 31, 66]. This is a noteworthy finding, given that CCT is focused primarily on compassion skills. However, mindfulness plays an important role as a foundation of compassion and prosocial practices [73]. For instance, mindfulness is formally trained in the early sessions of CCT as a foundation for subsequent practices [74]. Future studies should compare the differential effect of mindfulness and compassion-based programs in this population.

In line with previous studies examining the effects of CBIs [27, 37, 60, 66], there was a significant reduction of most maladaptive emotion-regulation strategies after the CCT (i.e., emotional inattention, emotional confusion, emotional rejection, and lack of emotional control). These skills are crucial in medical population, continually exposed to the suffering of their patients, which generates intense emotions that should be regulated, instead of ignoring and accumulating them, as this increases the risk of developing mental-health problems [2, 75]. Furthermore, we also found a significant decrease in concerns involving COVID-19 impact in the CCT group at follow-up, which is an important feature considering the reported negative impact of COVID-19 pandemic on clinicians' mental health [76].

Finally, to further test the efficacy of CCT, we used the RCI as a means of improving the individual-level analysis and the detection of potential adverse effects. The RCI revealed differences between CCT and WL in terms of clinically significant changes in stress, anxiety, emotional exhaustion, and cynicism. Generally, CCT showed higher recovery and improved rates, whereas WL showed higher number of participants with no changes or deteriorated. It bears noting that only a tiny fraction of participants showed a deterioration from baseline in CCT, which is an important result considering the potential adverse effects of meditation practice, such as increases in anxiety, depression or negative thinking [77].

## Strengths, limitations, and future directions

Our study shows certain strengths in that we have tried to overcome major limitations in this field [78], implementing some of the main recommendations for compassion interventions [25]. That is, the present study is a pilot randomized control trial with a waitlist control group, following CONSORT statements for pilot trials. Another noteworthy strength of this study was the inclusion of a follow-up assessment, an exception more than a rule in empirical compassion studies. Furthermore, although CCT was originally designed as an in-person program, we found that the positive effects remains despite being offered as an online intervention in the context of the second wave of COVID-19 pandemic.

Our study shows certain limitations. First, as a pilot study, the sample size was moderate, although it is in line with previous studies on CCT in medical students [32]. Since the final sample comprised fewer participants than expected, that may have reduced the statistical power of the trial. Second, we used a waitlist control group instead of an active control comparison. Third, the nature of CCT precludes blind participants from the intervention. Finally, we included only self-reported measures, although this is the most common and validated method for measuring psychological variables. Future studies should replicate our results with a larger sample size, an active control condition (e.g., a mindfulness training or a relaxation program), and measurements of biological variables (e.g., cortisol levels, heart-rate variability, and brain activity).

In view of the above, future studies should replicate our results with a larger sample size to extrapolate information about the efficacy of the program, an active control condition (e.g., a mindfulness training or a relaxation program), and measurements of biological variables (e.g., cortisol levels, heart-rate variability, and brain activity). Furthermore, some post-intervention changes persisted at follow-up (i.e., self-kindness, observing, non-reactivity, anxiety, emotional exhaustion, and some emotional-regulation factors), whereas others had vanished. Thus, future research should include longer longitudinal designs (e.g., one-year follow-up) while exploring the role of continued meditation practice after completion of the program as a mediator of follow-up effects. Future research should also compare our results with other compassion-based interventions (e.g., Compassion Focused Therapy), in order to examine whether other compassion programs could enhance the measures that did not improve with CCT. It would be informative also to analyze the effectiveness of the program in physicians working at hospitals as well as in medical students diagnosed with anxiety, depression, and/or burnout syndrome. Furthermore, given that universities are also responsible for sustainable human-resource development, these kinds of programs could also be applied to other university employees in order to improve their skills related to well-being. Finally, there could be cross-cultural differences in the importance of compassion in the medical curriculum, so futures studies should explore these potential differences.

## Conclusions

For all the above, the CCT appears to be an effective intervention to enhance compassion skills in a profession where compassion is such a vital and fundamental attribute. In addition, CCT also proves to be an effective intervention to reduce stress, anxiety, depression, and burnout symptoms, while reducing the maladaptive emotion-regulation strategies in a population particularly vulnerable to developing mental-health problems. This makes the CCT an excellent program to be included in hospital and medical-school training. Medical schools should aim not only for academic excellence, but also for training healthier and more compassionate professionals.

## Supporting information

**S1 Checklist. CONSORT 2010 checklist of information to include when reporting a randomised trial\*.**
(DOC)

**S1 File.**
(DOCX)

**S2 File.**
(PDF)

**S3 File.**
(PDF)

**S4 File.**
(PDF)

## Acknowledgments

The present work is part of an Innovation Teaching Program at Complutense University (2020-2021/139) and partially supported by Nirakara-lab. The authors want to thank all participants for their generosity in voluntarily participate in the study. We also thank Lilly Foundation, Maria Teresa García Antón, and Elena María Vara Ameigeiras for their help and inspiration throughout the project.

## Author Contributions

**Conceptualization:** Blanca Rojas, Pablo Roca.

**Data curation:** Elena Catalan, Pablo Roca.

**Formal analysis:** Pablo Roca.

**Funding acquisition:** Gustavo Diez.

**Investigation:** Blanca Rojas, Elena Catalan.

**Methodology:** Pablo Roca.

**Project administration:** Blanca Rojas.

**Resources:** Blanca Rojas.

**Supervision:** Blanca Rojas, Pablo Roca.

**Visualization:** Pablo Roca.

**Writing – original draft:** Blanca Rojas, Elena Catalan, Pablo Roca.

**Writing – review & editing:** Blanca Rojas, Gustavo Diez, Pablo Roca.

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
