## [Decision Letter · Decision Letter 0]

11 Apr 2023

PONE-D-23-02561A compassion-based program to reduce psychological distress in medical students: a pilot randomized clinical trialPLOS ONE

Dear Dr. Roca,

Thank you for submitting your manuscript to PLOS ONE. After careful consideration, we feel that it has merit but does not fully meet PLOS ONE’s publication criteria as it currently stands. Therefore, we invite you to submit a revised version of the manuscript that addresses the points raised during the review process.

The manuscript has been evaluated by three reviewers, and their comments are available below. The reviewers appreciate the importance of your study, but also raised a number of concerns, such as clarification on methodological aspects of the study and the interpretation of the results. Could you please revise the manuscript to carefully address the concerns raised?

We look forward to receiving your revised manuscript.

Kind regards,

Dario Ummarino, PhD

Senior Editor

PLOS ONE

2. We note that you have referenced (Jinpa T. Compassion cultivation training (CCT): Instructor’s manual. Unpublished, Stanford, CA. . 2010.) which has currently not yet been accepted for publication. Please remove this from your References and amend this to state in the body of your manuscript: (ie “Bewick et al. [Unpublished]”) as detailed online in our guide for authors

Reviewers' comments:

Reviewer's Responses to Questions

**Comments to the Author**

1. Is the manuscript technically sound, and do the data support the conclusions?

Reviewer #1: Yes

Reviewer #2: Yes

Reviewer #3: Yes

2. Has the statistical analysis been performed appropriately and rigorously? 

Reviewer #1: I Don't Know

Reviewer #2: Yes

Reviewer #3: Yes

3. Have the authors made all data underlying the findings in their manuscript fully available?

Reviewer #1: Yes

Reviewer #2: Yes

Reviewer #3: Yes

4. Is the manuscript presented in an intelligible fashion and written in standard English?

Reviewer #1: Yes

Reviewer #2: Yes

Reviewer #3: Yes

5. Review Comments to the Author

Reviewer #1: Important note: This review pertains only to ‘statistical aspects’ of the study and so ‘clinical aspects’ [like medical importance, relevance of the study, ‘clinical significance and implication(s)’ of the whole study, etc.] are to be evaluated [should be assessed] separately/independently. Further please note that any ‘statistical review’ is generally done under the assumption that (such) study specific methodological [as well as execution] issues are perfectly taken care of by the investigator(s). This review is not an exception to that and so does not cover clinical aspects {however, seldom comments are made only if those issues are intimately / scientifically related & intermingle with ‘statistical aspects’ of the study}. Agreed that ‘statistical methods’ are used as just tools here, however, they are vital part of methodology [and so should be given due importance]. I look at the manuscript in/with statistical view point, other reviewer(s) look(s) at it with different angle so that in totality the review is very comprehensive. However, there should be efforts from authors side to improve (may be by taking clues from reviewer’s comments). Therefore, please do not limit the revision only (with respect) to comments made here.

COMMENTS: I noted (i.e., in my opinion) that your ABSTRACT is well drafted, but is ‘assay type’. It is preferable [refer to item 1b of CONSORT checklist 2010: Structured summary of trial design, methods, results, and conclusions] to divide the ABSTRACT with small sections like ‘Objective(s)’, ‘Methods’, ‘Results’, ‘Conclusions’, etc. which is an accepted practice of most of the good/standard journals [including this one, though ‘The PLoS One Guidelines to Authors’ did not specify an Abstract format, it is desirable]. It will definitely be more informative then, I guess, whatever the article type may be [including a pilot randomized clinical trial].

A very small correction: meaning/longform of CCT [Compassion Cultivation Training] is given in the ‘abstract’ itself but on second time use of the term, whereas, it is a general convention to give it at first time use of the term. Of course, it a very minor correction suggested. It is stated in ‘abstract’ that “This pilot study aims to examine the effectiveness of Compassion-Based Interventions (CBIs) in preventing psychological distress” [is not ‘Interventions’ plural?] but next sentence “We hypothesize that the CCT program, as compared to a waitlist control group” indicates that this study deals with only one intervention namely CCT program. Is not that so? It is pardonable if it is by oversight/typing mistake, however, please clarify.

It seems that this ‘Clinical Trial’ is not register as desired. If it is, then why the number {ClinicalTrial.org (ID: XXX)} is not specified? This study being ‘pilot’ in nature, sample size is not a big issue. However, [though many things are ignored (loosely looked at / evaluated)] in case of ‘pilot studies’, methodological issues need to be very rigorous followed {like in case of clinical trial, CONSORT guidelines are to be strictly observed/followed}. Then the term ‘CONSORT’ is a very vital term [but surprisingly it appears only once in ‘Strengths, limitations, and future directions’ section]. Even important items of/in CONSORT checklist are not found [since your article type is ‘Clinical Trial’, you are supposed to cover these items in the report (& note that CONSORT for Pilot trial is/are also available).

According to document on CONSORT for Pilot trial “Formal hypothesis testing for effectiveness (or efficacy) is not recommended. The aim of a pilot trial is not to assess effectiveness (or efficacy) and it will usually be underpowered to do this” (check your aim stated). In addition, please note that any regression techniques [including Mixed-effects models which are conducted here to analyze the effects of the CCT program] are not basically/originally developed for any sort of [between or within group(s)] comparison(s). Using ‘Mixed-effects models’ is definitely not wrong; however, some head-to-head comparison is expected. Further, it may be noted that

Though the measures/tools used are appropriate [refer table-2], most of them are likely to yield data that are in ‘ordinal’ level of measurement [and not in ratio level of measurement for sure {as the score two times higher does not indicate presence of that parameter/phenomenon as double (for example, a Visual Analogue Scales VAS score or say ‘depression’ score)}]. Then application of suitable non-parametric test(s) is/are indicated/advisable [even if distribution may be ‘Gaussian’ (also called ‘normal’)]. Agreed that there is/are no non-parametric test(s)/technique(s) available to be used as alternative in all situation(s) [suitable / most desired/applicable], but should be used whenever/wherever they are available. Therefore, in short use suitable non-parametric test(s)/technique(s) while dealing with data that are in ‘ordinal’ level of measurement even if [despite that] the distribution may be ‘Gaussian’. Testing ‘normality’ in sample [by using any normality test(s)} is not required/desired while dealing with data that are in ‘ordinal’ level of measurement [as most of the normality tests are not valid for ‘ordinal’ data].

I request authors to check contents of references 77,78. Because in my knowledge, at least article 77 concludes as follows:

Conclusions: Engagement in MBSR is not predictive of increased rates of harm relative to no treatment. Rather, MBSR may be protective against multiple indices of harm. Research characterizing the relatively small proportion of MBSR participants that experience harm remains important.

Moreover, it may please be noted that “Absence of evidence is not evidence of absence” [Altman DG, Bland JM. BMJ volume 311, 1995, p 485 (Reprinted: Australian Veterinary Journal 1996;74, 311)]. {Even when P-value is not significantly lower that is null hypothesis of no difference / no association is not rejected, (in short, result is not significant), that does not amount to evidence of absence i.e., it does not imply that there no difference / no association. It only implies that there is no (i.e., these samples do not provide) [say enough] evidence to prove (rather indicate with certain specified confidence level) the difference / association}. Therefore, conclusion(s) from any study [in which result(s) is/are not significant], should be drawn in the light of this fact. Particularly look at (due consideration of) the hypothesis in each of the study quoted in reference 78.

Although the ‘Intervention description’ is indeed very good and useful, [as pointed out in ‘important note’ above] note that “This review pertains only to ‘statistical aspects’ of the study and so ‘clinical aspects’ should be assessed separately/independently [one should carefully consider/look at the clinical implications of the study].

In my opinion, to rescue this article (which is quite possible and easy), a small amount of re-vision (re-drafting) may be needed. However, please do not limit the revision only (with respect) to comments made here. More improvement is expected. Recommending minor revision.

Reviewer #2: Thanks for allowing me to review the present manuscript. This is an interesting research report on the effectiveness of a Compassion-Based Intervention (CCT) in reducing distress in medical students. The study is a randomized-controlled trial. I think this is an important topic and a well-written manuscript that should be accepted for publication.

Reviewer #3: “A compassion-based program to reduce psychological distress in medical students: a

pilot randomized clinical trial”

The present study aims to investigate the impact of Compassion Cultivation Training on psychological outcomes in a sample of medical students. The topic is exciting, and the contribution that the current study could provide to scientific literature is influential and essential.

Following this, I reported a series of suggestions to help authors improve the manuscript.

Background:

Since there are other types of Protocol Intervention, CBT-oriented focused on increases in

“Compassion, empathy and mindfulness skills, reductions of

psychological distress (stress, anxiety, and depression) and burnout, improvements in

well-being and emotion regulation, among others” I suggest that authors explain better why they chose this specific training and not others.

Materials and Methods:

The number of the study protocol approved by the Ethical Committee and the pre-registration at ClinicalTrial.org need to be included.

Participants:

I kindly ask the authors to specify the background related to the decision to set a medium effect size to estimate sample size, also considering that this field is novel and not extensively investigated, it should be expected to have a sample composed of a higher number of participants to extrapolate information about efficacy in a pilot study.

Discussion:

Since the final sample comprised fewer participants than expected, I suggest that authors better explain the implication of this fact on the outcome interpretation. I suggest adding this part to the conclusion section.

6. PLOS authors have the option to publish the peer review history of their article (what does this mean?). If published, this will include your full peer review and any attached files.

Reviewer #1: No

Reviewer #2: No

Reviewer #3: **Yes: **Susanna Pardini

---

## [Author Response · Author response to Decision Letter 0]

29 May 2023

We thank the reviewers for their constructive comments and suggestions that have markedly improved our manuscript. Please see below our responses to each comment.

Reviewer #1

1.1. I noted (i.e., in my opinion) that your ABSTRACT is well drafted, but is ‘assay type’. It is preferable [refer to item 1b of CONSORT checklist 2010: Structured summary of trial design, methods, results, and conclusions] to divide the ABSTRACT with small sections like ‘Objective(s)’, ‘Methods’, ‘Results’, ‘Conclusions’, etc. which is an accepted practice of most of the good/standard journals [including this one, though ‘The PLoS One Guidelines to Authors’ did not specify an Abstract format, it is desirable]. It will definitely be more informative then, I guess, whatever the article type may be [including a pilot randomized clinical trial].

We thank the reviewer for this suggestion. We have adapted the abstract according to the item 1b of the CONSORT checklist 2010, which is also reflected in the CONSORT checklist for pilot studies (see point 1.5 below). 

1.2. A very small correction: meaning/longform of CCT [Compassion Cultivation Training] is given in the ‘abstract’ itself but on second time use of the term, whereas, it is a general convention to give it at first time use of the term. Of course, it a very minor correction suggested.

Done!

1.3. It is stated in ‘abstract’ that “This pilot study aims to examine the effectiveness of Compassion-Based Interventions (CBIs) in preventing psychological distress” [is not ‘Interventions’ plural?] but next sentence “We hypothesize that the CCT program, as compared to a waitlist control group” indicates that this study deals with only one intervention namely CCT program. Is not that so? It is pardonable if it is by oversight/typing mistake, however, please clarify. 

We thank the reviewer for this observation. Indeed, the CCT is one of the many Compassion-Based Interventions that have been developed. We selected it because it is one of the programs with the most empirical evidence in this field. To avoid any confusion in the abstract, we have replaced CBIs with the CCT (p. 2), which is the program we have evaluated in this study. 

1.4. It seems that this ‘Clinical Trial’ is not register as desired. If it is, then why the number {ClinicalTrial.org (ID: XXX)} is not specified?

Clinical Trial ID was intentionally hidden to maintain the anonymity during the review process. The ID (NCT04690452) is now included in the Study Design section (p. 6). 

1.5. This study being ‘pilot’ in nature, sample size is not a big issue. However, [though many things are ignored (loosely looked at / evaluated)] in case of ‘pilot studies’, methodological issues need to be very rigorous followed {like in case of clinical trial, CONSORT guidelines are to be strictly observed/followed}. Then the term ‘CONSORT’ is a very vital term [but surprisingly it appears only once in ‘Strengths, limitations, and future directions’ section]. Even important items of/in CONSORT checklist are not found [since your article type is ‘Clinical Trial’, you are supposed to cover these items in the report (& note that CONSORT for Pilot trial is/are also available).

We thank the reviewer for rising this. The CONSORT checklist for pilot studies (Eldridge et al., 2016) is now available as a table in the Supplementary Materials. Furthermore, the adhesion to the CONSORT recommendations for pilot studies is now mentioned in the Study Design (p. 5) and Discussion (p. 19) sections.

6.1. According to document on CONSORT for Pilot trial “Formal hypothesis testing for effectiveness (or efficacy) is not recommended. The aim of a pilot trial is not to assess effectiveness (or efficacy) and it will usually be underpowered to do this” (check your aim stated). 

Following CONSORT recommendations, we have changed “effectiveness” to “feasibility” as the aim of the study (p. 2).

7.1. In addition, please note that any regression techniques [including Mixed-effects models which are conducted here to analyze the effects of the CCT program] are not basically/originally developed for any sort of [between or within group(s)] comparison(s). Using ‘Mixed-effects models’ is definitely not wrong; however, some head-to-head comparison is expected. 

We thank the reviewer for this suggestion. As can be seen in Table 2, we analyzed between and within groups differences, including Group and Time main effects and the interactions. Furthermore, we computed post-hoc comparisons (adjusted by Tukey) to determine which Group X Time interactions are responsible for the significant differences. However, we neglected to mention these comparisons in the Data Analysis plan, and they were described in little detail in the results. Therefore, we have improved the explanation in Data Analysis (p. 10) and Results (p. 11) sections.

8.1. Though the measures/tools used are appropriate [refer table-2], most of them are likely to yield data that are in ‘ordinal’ level of measurement [and not in ratio level of measurement for sure {as the score two times higher does not indicate presence of that parameter/phenomenon as double (for example, a Visual Analogue Scales VAS score or say ‘depression’ score)}]. Then application of suitable non-parametric test(s) is/are indicated/advisable [even if distribution may be ‘Gaussian’ (also called ‘normal’)]. Agreed that there is/are no non-parametric test(s)/technique(s) available to be used as alternative in all situation(s) [suitable / most desired/applicable], but should be used whenever/wherever they are available. Therefore, in short use suitable non-parametric test(s)/technique(s) while dealing with data that are in ‘ordinal’ level of measurement even if [despite that] the distribution may be ‘Gaussian’. Testing ‘normality’ in sample [by using any normality test(s)} is not required/desired while dealing with data that are in ‘ordinal’ level of measurement [as most of the normality tests are not valid for ‘ordinal’ data].

Although the questionnaires used in our study are ordinal at the item level, they are quantitative at the factor level (i.e., sum scores), which makes them suitable to be analyzed by means of parametric tests. This is a very common practice in areas such as psychology, where they use procedures very similar to those used in our manuscript (e.g., González-Robles et al., 2022; Romero-Ferreiro et al., 2022). 

González-Robles, A., Roca, P., Díaz-García, A., García-Palacios, A., & Botella, C. (2022). Long-term Effectiveness and Predictors of Transdiagnostic Internet-Delivered Cognitive Behavioral Therapy for Emotional Disorders in Specialized Care: Secondary Analysis of a Randomized Controlled Trial. JMIR Mental Health, 9(10), e40268.

Romero-Ferreiro, V., García-Fernández, L., Aparicio, A. I., Martínez-Gras, I., Dompablo, M., Sánchez-Pastor, L., ... & Rodriguez-Jimenez, R. (2022). Emotional Processing Profile in Patients with First Episode Schizophrenia: The Influence of Neurocognition. Journal of Clinical Medicine, 11(7), 2044.

9.1. I request authors to check contents of references 77,78. Because in my knowledge, at least article 77 concludes as follows: “Conclusions: Engagement in MBSR is not predictive of increased rates of harm relative to no treatment. Rather, MBSR may be protective against multiple indices of harm. Research characterizing the relatively small proportion of MBSR participants that experience harm remains important”. Moreover, it may please be noted that “Absence of evidence is not evidence of absence” [Altman DG, Bland JM. BMJ volume 311, 1995, p 485 (Reprinted: Australian Veterinary Journal 1996;74, 311)]. {Even when P-value is not significantly lower that is null hypothesis of no difference / no association is not rejected, (in short, result is not significant), that does not amount to evidence of absence i.e., it does not imply that there no difference / no association. It only implies that there is no (i.e., these samples do not provide) [say enough] evidence to prove (rather indicate with certain specified confidence level) the difference / association}. Therefore, conclusion(s) from any study [in which result(s) is/are not significant], should be drawn in the light of this fact. Particularly look at (due consideration of) the hypothesis in each of the study quoted in reference 78.

We thank the reviewer for this observation. We used reference 77 as an example of the debate about the potential adverse effects of meditation programs, which is well covered in its introduction and discussion. However, we agree with the reviewer that it might be confusing given its results. Therefore, we have replaced these references with Baer et al. (2019) study, which best fits the content of the sentence (p. 19).

Baer, R., Crane, C., Miller, E., & Kuyken, W. (2019). Doing no harm in mindfulness-based programs: conceptual issues and empirical findings. Clinical psychology review, 71, 101-114.

10.1 Although the ‘Intervention description’ is indeed very good and useful, [as pointed out in ‘important note’ above] note that “This review pertains only to ‘statistical aspects’ of the study and so ‘clinical aspects’ should be assessed separately/independently [one should carefully consider/look at the clinical implications of the study]. In my opinion, to rescue this article (which is quite possible and easy), a small amount of re-vision (re-drafting) may be needed. However, please do not limit the revision only (with respect) to comments made here. More improvement is expected. Recommending minor revision.

Once again, many thanks to the reviewer for these very helpful comments and suggestions!

Reviewer #2:

Thanks for allowing me to review the present manuscript. This is an interesting research report on the effectiveness of a Compassion-Based Intervention (CCT) in reducing distress in medical students. The study is a randomized-controlled trial. I think this is an important topic and a well-written manuscript that should be accepted for publication.

We are glad that the reviewer enjoyed the manuscript and recognized the importance of the topic it addresses.

Reviewer #3

The present study aims to investigate the impact of Compassion Cultivation Training on psychological outcomes in a sample of medical students. The topic is exciting, and the contribution that the current study could provide to scientific literature is influential and essential. Following this, I reported a series of suggestions to help authors improve the manuscript.

3.1. Since there are other types of Protocol Intervention, CBT-oriented focused on increases in “Compassion, empathy and mindfulness skills, reductions of psychological distress (stress, anxiety, and depression) and burnout, improvements in well-being and emotion regulation, among others” I suggest that authors explain better why they chose this specific training and not others.

We thank the reviewer for this suggestion. There are several reasons to explain why we chose CCT among all compassion-based programs: 1) CCT has shown promising results in improving the targets that we were interested in (e.g., psychological distress, emotion regulation, mindfulness…); 2) the CCT was developed at Stanford University with a duration and format well adapted to university settings. Furthermore, the CCT have shown promising results in medical students (Weingartner et al., 2019); and 3) a member of our research team is certified by the Compassion Institute to apply the CCT, which enhanced the feasibility of the project. We have now explained these reasons in more detail in the Introduction (p. 4) and Method (p. 8) sections. 

3.2. The number of the study protocol approved by the Ethical Committee and the pre-registration at ClinicalTrial.org need to be included.

We thank the reviewer for this observation. Clinical Trial ID and the Ethical Committee ID were intentionally hidden to maintain the anonymity during the review process. Both are now included in the Study Design section (p. 6). 

3.3. I kindly ask the authors to specify the background related to the decision to set a medium effect size to estimate sample size, also considering that this field is novel and not extensively investigated, it should be expected to have a sample composed of a higher number of participants to extrapolate information about efficacy in a pilot study.

We thank the reviewer for the suggestion. We used a medium effect size because previous studies using the CCT have found effects around .40 in the same variables that we are using in our study (e.g., Brito-Pons et al., 2018; Roca et al., 2021). Furthermore, as mentioned by Reviewer #1 in point 1.5, our study is pilot in nature, so sample size is not a major concern. However, we have now explained better this rationale in the Method (p. 6), including a reflection in the Discussion on the need to increase the sample size in future studies to extrapolate information about efficacy of the program (p. 20). 

3.4. Since the final sample comprised fewer participants than expected, I suggest that authors better explain the implication of this fact on the outcome interpretation. I suggest adding this part to the conclusion section.

Following the reviewer’s suggestion, we have included a limitation in the Discussion section about how the sample size may have reduced the statistical power of the trial (p. 20).

---

We thank you for your constructive comments that have markedly improved our manuscript.

---

## [Decision Letter · Decision Letter 1]

5 Jun 2023

A compassion-based program to reduce psychological distress in medical students: a pilot randomized clinical trial

PONE-D-23-02561R1

Dear Dr. Roca,

We’re pleased to inform you that your manuscript has been judged scientifically suitable for publication and will be formally accepted for publication once it meets all outstanding technical requirements.

Kind regards,

Samuel Yeung-shan Wong

Academic Editor

PLOS ONE

Additional Editor Comments (optional):

Reviewers' comments:

Reviewer's Responses to Questions

**Comments to the Author**

1. If the authors have adequately addressed your comments raised in a previous round of review and you feel that this manuscript is now acceptable for publication, you may indicate that here to bypass the “Comments to the Author” section, enter your conflict of interest statement in the “Confidential to Editor” section, and submit your "Accept" recommendation.

Reviewer #2: All comments have been addressed

Reviewer #3: All comments have been addressed

2. Is the manuscript technically sound, and do the data support the conclusions?

Reviewer #2: Yes

Reviewer #3: Yes

3. Has the statistical analysis been performed appropriately and rigorously? 

Reviewer #2: Yes

Reviewer #3: Yes

4. Have the authors made all data underlying the findings in their manuscript fully available?

Reviewer #2: Yes

Reviewer #3: Yes

5. Is the manuscript presented in an intelligible fashion and written in standard English?

Reviewer #2: Yes

Reviewer #3: Yes

6. Review Comments to the Author

Reviewer #2: The authors have adequately addressed my comments raised in the previous round of review and I feel that this manuscript is acceptable for publication.

Reviewer #3: (No Response)

7. PLOS authors have the option to publish the peer review history of their article (what does this mean?). If published, this will include your full peer review and any attached files.

Reviewer #2: No

Reviewer #3: **Yes: **Susanna Pardini

---

## [Editor Report · Acceptance letter]

12 Jun 2023

PONE-D-23-02561R1 

A compassion-based program to reduce psychological distress in medical students: a pilot randomized clinical trial. 

Dear Dr. Roca:

I'm pleased to inform you that your manuscript has been deemed suitable for publication in PLOS ONE. Congratulations! Your manuscript is now with our production department. 

Kind regards, 

on behalf of

Dr. Samuel Yeung-shan Wong 

Academic Editor

PLOS ONE